# A Gene-Based Method for Cytogenetic Mapping of Repeat-Rich Mosquito Genomes

**DOI:** 10.3390/insects12020138

**Published:** 2021-02-06

**Authors:** Reem A. Masri, Dmitriy A. Karagodin, Atashi Sharma, Maria V. Sharakhova

**Affiliations:** 1Department of Entomology and the Fralin Life Sciences Institute, Virginia Polytechnic Institute and State University, Blacksburg, VA 24061, USA; reemm7@vt.edu; 2Laboratory of Evolutionary Genomics of Insects, The Federal Research Center Institute of Cytology and Genetics, Siberian Branch of the Russian Academy of Sciences, 630090 Novosibirsk, Russia; karagodin@bionet.nsc.ru; 3Department of Biochemistry and the Fralin Life Sciences Institute, Virginia Polytechnic Institute and State University, Blacksburg, VA 24061, USA; atashi04@vt.edu

**Keywords:** genome, physical mapping, DNA hybridization

## Abstract

**Simple Summary:**

Mosquitoes from the family Culicidae are vectors of numerous human diseases such as dengue, Zika, and West Nile fevers, malaria, and lymphatic filariasis. The availability of high-quality genome assemblies for mosquitoes assists in identifying genes responsible for important epidemiological traits such as vector competence, insecticide resistance, and mosquito behavior and stimulates a better understanding of the genetic structure of mosquito populations. Thus, the development of genomic resources and tools is a necessary step for designing and implementing novel genome-based approaches to control mosquitoes. The emphasis of this study is the further development and optimization of a physical genome mapping approach. Physical mapping is placing genomic scaffolds to the chromosomes based on hybridization of the specific probes. This task is difficult for mosquitoes with large genome sizes which are enriched with repetitive DNA sequences and can be potentially misassembled. Here, we describe in detail a simple and robust technique for the physical mapping of such mosquito genomes. This method can be further used for physical genome mapping in other mosquitoes and insects.

**Abstract:**

Long-read sequencing technologies have opened up new avenues of research on the mosquito genome biology, enabling scientists to better understand the remarkable abilities of vectors for transmitting pathogens. Although new genome mapping technologies such as Hi-C scaffolding and optical mapping may significantly improve the quality of genomes, only cytogenetic mapping, with the help of fluorescence in situ hybridization (FISH), connects genomic scaffolds to a particular chromosome and chromosome band. This mapping approach is important for creating and validating chromosome-scale genome assemblies for mosquitoes with repeat-rich genomes, which can potentially be misassembled. In this study, we describe a new gene-based physical mapping approach that was optimized using the newly assembled *Aedes albopictus* genome, which is enriched with transposable elements. To avoid amplification of the repetitive DNA, 15 protein-coding gene transcripts were used for the probe design. Instead of using genomic DNA, complementary DNA was utilized as a template for development of the PCR-amplified probes for FISH. All probes were successfully amplified and mapped to specific chromosome bands. The genome-unique probes allowed to perform unambiguous mapping of genomic scaffolds to chromosome regions. The method described in detail here can be used for physical genome mapping in other insects.

## 1. Introduction

Mosquito genomics is rapidly developing as a discipline that can answer a large spectrum of biological questions. However, its major focus is assisting in the development of new strategies for mosquito control [1]. Mosquito genomics is employed to address new threats in the spread of infectious diseases such as malaria, dengue, Zika fever, chikungunya, and others [2,3]. With the help of novel and affordable sequencing technologies such as long-read technologies from Pacific Biosciences (PacBio) [4] and Oxford Nanopore Technology (ONT) [5,6], the number of mosquito genomes available through public databases is rapidly increasing. Nevertheless, a high number of mosquito genomes are still far from complete and are represented by multiple relatively short scaffolds with low N50 statistics; only a few mosquito genomes are assembled at the chromosome-scale level. Most of the mosquitoes with chromosome-level assemblies belong to malaria vectors from the subfamily Anophelinae: *Anopheles gambiae* [7,8,9], *An. albimanus* [10,11,12], *An. atroparvus* [12,13,14], *An. stephensi* [12,14], *An. funestus* [14,15], *An. merus* [12], *An. arabiensis* [16], and *An. coluzzii* [16]. Only one chromosome-scale assembly has been developed for Culicinae mosquitoes, the major vector of arboviruses, *Aedes aegypti* [17]. Recently, a significantly improved version of the Asian tiger mosquito *Ae. albopictus* genome became available [18]. This effort placed 57% of the genome to the chromosomes. However, chromosome-level genome assembly for this species is still missing.

There are two major components that affect the quality of the genome: heterozygosity and repetitive DNA sequences. The first problem, heterozygosity, can be avoided by using inbred homozygous individuals [17] or by sequencing a single mosquito genome [19]. The second problem, repetitive DNA sequences, leads to algorithmically compressed sequences [20]. Repetitive DNA sequences are a significant impediment for genome quality in mosquitoes from the Culicinae subfamily [21,22]. Unlike Anophelinae mosquito genomes, Culicinae genomes are extremely enriched by different classes of transposable elements which have a tendency to localize in gene introns and spread throughout the genome [23,24]. This problem can be resolved through additional direct mapping of the genomic scaffolds to the chromosomes. Although new physical mapping technologies such as optical mapping [25] or Hi-C scaffolding [26,27] could significantly increase contiguity of the genomic scaffolds, only cytogenetic mapping based on fluorescence in situ hybridization (FISH) provides physical evidence of the scaffold location in particular regions of the chromosomes [28]. This method allows one to relate DNA probes of a known sequence to a particular band on the chromosomes. Thus, further development of the cytogenetic physical method is crucial. 

The goal of the current study is to simplify and further optimize the cytogenetic mapping technique for highly repetitive mosquito genomes. We propose a new approach based on PCR amplification of DNA probes using complementary DNA (cDNA) that does not include repetitive DNA sequences from gene introns. This method was used for the development of a physical map for *Ae. albopictus* and is based on the hybridization of fifty cDNA fragments or gene exons from twenty-four scaffolds to mitotic chromosomes from imaginal discs, generating the first physical map of the *Ae. albopictus* genome [18]. The genome of this mosquito is enriched with transposable elements, making its size between 1.190 and 1.275 Gb long across populations, which is around five times larger than the genome of the major malaria vector *An. gambiae* [7]. Here, we describe this method in all required details so that it can be applied towards physical genome mapping in other mosquitoes or in other insect species of interest. 

## 2. Materials and Methods 

In this study, we used the Foshan strain, which was utilized for sequencing of the first and the last versions of the *Ae. albopictus* genome [18,29]. Mosquito eggs were hatched at 28 °C, and after several days, 2nd or 3rd instar larvae were transferred to 20 °C to obtain a high number of mitotic divisions in the imaginal discs of the 4th instar larvae.

At the time when the experiments were planned, the *Ae. albopictus* Aalbo_primary.1 genome assembly was still unavailable [19]. Therefore, we used a transcriptome from the C6/36 cell line of *Ae. albopictus* [30] to design target probes for FISH (Figure 1A). A total of 15 transcripts which represent protein-coding genes with unique locations in the newly assembled genome, with sizes bigger than 5000 nucleotides (nt), were selected (Table 1). Primers were developed using the primer3plus program [31] with the following criteria: optimal primer size of 23 nt, melting temperature (Tm of 58 °C with maximum Tm difference of 5 °C, and GC content of 45%. The melting temperatures of the oligonucleotides were determined using the New England Biolab’s Tm calculator (NEB, New England Biolabs, Ipswich, MA, USA). To visually see the FISH signal on mitotic chromosomes using microscopy, the final sizes of the PCR products were at least 3800 nt.

For cDNA synthesis, RNA was first extracted from ten 24 h old gravid female mosquito ovaries (Figure 1B). Female mosquitoes were placed on ice for immobilization; then, the ovaries were dissected in distilled water using a binocular microscope and immediately placed in TRIzol reagent (Thermo Fisher Scientific Inc., Waltham, MA, USA). Extraction was then performed using the Direct-zol RNA Miniprep kit (Zymoresearch, Irvine, CA, USA) following the manufacturer’s protocol. cDNA was synthesized using ~200 ng RNA and Superscript III Reverse Transcriptase following the manufacturer’s protocol (Thermo Fisher Scientific Inc., Waltham, MA, USA). cDNA synthesis was performed using either oligo(dT), random hexamer, or gene-specific primers. However, after multiple attempts and troubleshooting steps, cDNA synthesized using oligo(dT) was determined to be the only successful method to give the desired band on agarose gels. RNA residuals were digested following the manufacturer’s protocol by adding 1 µL of RNase H to each tube and incubating for 20 min at 37 °C. For later use, cDNA was diluted in Fisher DNAse/RNAse free water (Thermo Fisher Scientific Inc., Waltham, MA, USA) to 100 ng/µL and stored at −20 °C.

To amplify DNA fragments, PCR was performed with the specific primers mentioned above (Table 1) using NEB Q5 high-fidelity DNA polymerase following the manufacturer’s protocol (New England Biolabs, Ipswich, MA, USA). Briefly, 10 µL of 5× Q5 reaction buffer, 1 µL 10mM dNTPs, 2.5 µL of 10 µM forward primer, 2.5 µL of 10 µM reverse primer, 800 ng of cDNA as a DNA template, 0.5 µL 2 u/µL Q5 high-fidelity DNA polymerase, 10 µL 5× Q5 high GC enhancer, and nuclease-free water for a 50 µL reaction were used. PCR cycle conditions following the protocol were: initial denaturation, 98 °C for 10 min; denaturation at 98 °C for 1 min; annealing at 50–72 °C for 30 s (optimal annealing temperature was 5 °C degrees below our lowest primer Tm); and 72 °C for 30 s per 1000 nt (2 min 30 s for an average 5000 nt probe). The optimal cycle number was 32. However, this number can be manipulated from 25 to 35 to obtain an optimal gel band. The final extension was at 72 °C for 2 min then held at 4 °C. Briefly, 5 µL of PCR product was run on 1% agarose to check for the desired DNA fragment (Figure 1B), and then, DNA was purified from the PCR mixture using Promega Wizard SV gel and a PCR clean-up system protocol (Promega Corporation, Madison, WI, USA).

For FISH, slides were prepared from imaginal discs of 4th instar larvae of *Ae. albopictus*. Before dissection, larvae were placed on ice for several minutes to immobilize them. Then, larvae were transferred to a slide along with a drop of a cold, hypotonic solution of 0.5% sodium citrate (Fisher scientific, Waltham, MA, USA) and dissected under an Olympus SZ microscope (Olympus America, Inc., Melville, NY, USA). The dissection procedure has been thoroughly explained in the previously published protocol [32]. Slides were analyzed under an Olympus CX41 microscope (Olympus America, Inc., Melville, NY, USA) at 400× magnification (40× objective and 10× eyepieces) (Figure 1C).

Purified DNA was labeled with Cy3- or Cy5-dUTP (Enzo Life Sciences Inc., Farmingdale, NY, USA) by nick translation using published protocols (Figure 1B), but probe incubation was performed for 1 h 45 min instead of 2 h 30 min [32]. The duration time was decreased to obtain the optimal fragment sizes between 300 and 500 nt. The increased time resulted in over-digestion of the probe, which reduced the effectiveness of FISH. The hybridization buffer which was used to dissolve the probes contained 50% formamide, 10% dextran sulfate sodium (DSS), and 0.1% Tween 20 in 2× Saline-Sodium Citrate, SSC (Thermo Fisher Scientific, Waltham, MA, USA). In situ hybridization (Figure 1C) was performed using a modified protocol [32]. Briefly, 1 µL of 10 mg/µL RNase A in 100 µL of 2× SSC solution was placed on each slide for 30 min. Slides were then placed in the following solutions: 2× SSC for 5 min at 37 °C, and pretreated in pepsin solution (50 mL sterile water, 50 µL 1M HCl, 0.1 mg/mL of pepsin) for 5 min at 37 °C. After that, slides were washed in 1× Phosphor Buffered Saline, PBS (Thermo Fisher Scientific, Waltham, MA, USA) and formaldehyde fixation solution (1.5 mL 37% formaldehyde to 50 mL 1× PBS) for 10 min each at room temperature. They were then rinsed in 1× PBS for 5 min and dehydrated using an ethanol series (70%, 80%, and 100%) for 5 min each and dried. Then, 250 ng of probe in hybridization buffer was added to the slide, covered with a 22 × 22-mm glass coverslip, and sealed with Elmer’s rubber cement. The slides were then placed on a Thermobrite (Abbott Molecular Inc., Chicago, IL, USA) programmer at 73 °C for 5 min, followed by 37 °C overnight. The next day, slides were washed using the following solutions: in 1× SSC, pre-warmed at 60 °C, for 5 min and, at 37 °C, 4× SSC/NP40 (40 mL of water, 10 mL of 20× SSC, and 50 µL of NP40) for 10 min. After that, slides were rinsed in 1× PBS and stained in 50 µL of Oxazol Yellow, YOYO-1 stain (Thermo Fisher Scientific, Waltham, MA, USA) at a 1:1000 dilution in 1× PBS under parafilm in a dark box for 15 min. Slides were then rinsed in 1× PBS for 3 min and a drop of prolong gold antifade (Invitrogen Corporation, Carlsbad, CA, USA) was added to each slide. Slides were covered with glass coverslips and analyzed using a Zeiss LSM 880 Confocal Microscope (Carl Zeiss Microimaging, Inc., Thornwood, NY, USA) at 1000× magnification (100× objective and 10× eyepieces) (Figure 1C).

## 3. Results

In this study, we describe in detail a new physical approach for mapping repeat-rich mosquito genomes to their mitotic chromosomes. This method was used, for the first time, to physically map the *Ae. albopictus* genome [18]. Here, we describe this technique with all the methodological details required for further utilization of this method. Figure 1 summarizes the entire procedure. For the probe design, we utilized transcripts from the *Ae. albopictus* cell line genome (Figure 1A) [30]. Based on our genome analyses, these transcripts have unique locations in the genome and, thus, can be used for further physical genome mapping (Table 1). To simplify primer design, the original sizes of the transcripts varied from 5012 to 10,567 bases. The primers were designed to obtain a final product of the DNA probe between 3800 and 5000 nt. The size of the DNA probe was chosen so as to obtain a clear signal on small mitotic chromosomes. Mosquito ovaries were utilized to obtain RNA for further development of a large amount of cDNA (Figure 1B). cDNA was produced from mRNA that mostly represents gene exons and does not contain gene introns, which are enriched with different classes of transposable elements in repeat-rich Culicinae genomes [23,33]. Repetitive DNA from gene introns produces multiple unspecific signals on the chromosomes that make it impossible to detect the correct position of a gene probe.

After the probes were designed, we used standard procedures [28] to obtain chromosome preparations from 4th instar imaginal discs of *Ae. albopictus* (Figure 1C) and proceeded with probe labeling and FISH (Figure 1C and Figure 2). Similar to other mosquitoes, the *Ae. albopictus* karyotype includes three pairs of chromosomes. In correspondence with *Ae. aegypti* chromosomes [34], the shortest chromosome was described as chromosome 1, the longest chromosome was identified as chromosome 2, and the mid-length chromosome was designated as chromosome 3 [18]. Unlike the anophelines, the sex-determining chromosomes in all culicine mosquitoes, including *Ae. albopictus*, are homomorphic [35]. For the physical mapping of the DNA probes (Figure 1C and Figure 3), we utilized previously developed chromosome idiograms [18]. Chromosomes of *Ae. albopictus* can easily be distinguished using chromosome and arm length differences [18]. Furthermore, *Ae. albopictus* chromosomes have a clear banding pattern that aids in physical mapping. In this study, 15 PCR-amplified probes and rDNA were mapped to the bands on an idiogram determined by FISH (Table 1; Figure 2 and Figure 3). An 18 S ribosomal DNA probe that hybridized to the secondary constriction on 1q22 was used as an additional marker for chromosome 1 (Figure 1C and Figure 3). The positions of the mapped transcripts were consistent with their positions in the *Ae. aegypti* genome, supporting the accuracy of the physical mapping.

## 4. Discussion

Cytogenetic genome mapping could utilize different sources of DNA probes and chromosomes, depending on the size of the genomes and the quality of the chromosomes. This study focuses on the further development of cytogenetic physical mapping for application in highly repetitive genome research in mosquitoes such as *Ae. albopictus*. A detailed comparison of the new gene-based method and the conventional Bacterial Artificial Chromosome (BAC)-based approach for physical genome mapping is summarized in Table 2. Although BAC clones have been successfully used for physical genome mapping for *An. gambiae* [7], *Ae. aegypti* [17,23], and *Cx. quinquefasciatus* [36], utilizing BAC clones for highly repetitive genomes is challenging. First, BAC library development and BAC-end sequencing are expensive and unproductive in repeat-rich genomes because a large portion of the BAC-ends are represented by repeats located in multiple places in the genome. Second, blocking unspecific BAC clone hybridization to chromosomes by unlabeled repeats, so-called Cot-1 DNA, is also an inefficient and extremely time-consuming process as it requires extraction of large amounts of genomic DNA. Utilizing cDNA as a template for the probe development, which we proposed in this study, allows for specific hybridization of the probes to the chromosomes without additional DNA sequencing and repeat blocking during FISH. Although both methods are labor-intensive, the gene-based method allows us to avoid long procedures such as extraction of multiple BAC clones. Using a PCR-based approach simplifies the probe design because it requires only a primer design and DNA amplification. The duration of the entire FISH procedure is reduced since a separate probe denaturation step is not required for the PCR-amplified probe.

Another challenge associated with physical mapping in mosquitoes with highly repetitive genomes is the poor quality of their polytene chromosomes. Thus, mitotic chromosomes became a major resource for cytogenetic mapping in Culicinae mosquitoes [35,36]. However, using mitotic chromosomes requires large DNA probes for FISH. For example, ~500 nt probes, which are clearly visible in FISH on large polytene chromosomes in malaria mosquito chromosomes [10,13], are not visible on small mitotic chromosomes. Thus, in our study, only probes with sizes larger than 3800 nt were utilized for FISH. Although an alternative method based on amplification of the gene exons for the FISH probes was also used for physical mapping of the *Ae. albopictus* genome [18], this method is not efficient for Culicinae mosquitoes because most of the exons in these mosquito genomes are smaller than 38 nt and would be invisible on mitotic chromosomes after FISH.

In addition to the cytogenetic approaches that we describe here, anchoring the genome assembly to the chromosomes can be performed by using different methods such as classical genetic linkage mapping [37,38,39] or by using modern technologies such as optical mapping [25] and Hi-C scaffolding [26,27]. Each of these mapping methods has its own advantages and limitations. For example, genetic linkage maps, which are based on recombination frequencies, offer the opportunity to connect genomic assemblies with epidemiologically important traits in mosquitoes, including their ability to transmit pathogens [17,40]. However, due to low recombination rates around the centromeres, genetic linkage maps have very low resolution around the centromeres [37,39]. New approaches such as optical mapping based on high-resolution restriction maps from single, stained molecules of DNA [25] or the Hi-C method, which uses the frequency of chromosome contacts to estimate the distance between DNA fragments [27], are highly efficient for linear ordering of genomic scaffolds at the molecular level. However, both methods are useless in connecting the genome assembly with particular chromosomes or chromosome structures such as telomeres, centromeres, and chromosome bands. Overall, we think that the best strategy for the development of high-quality genomes is to use a combination of different genome mapping techniques in one study. For example, one of the best chromosome-scale assemblies developed for mosquitoes, the *Ae. aegypti* genome, is based on a combination of modern technologies, Hi-C and optical mapping, with classical chromosome mapping technologies and linkage mapping [17].

## 5. Conclusions

High-resolution cytogenetic maps are fundamental for anchoring genome sequences onto chromosomes. Using complementary DNA as a probe template for FISH, we developed a new approach for the construction of physical maps and described this method in detail using the newly assembled *Ae. albopictus* genome as an example. This method is especially beneficial for the physical mapping of the highly repetitive Culicinae mosquito genome but can also be used for development of physical genome maps for other mosquitoes or insects to better aid in their genome assembly.

## Figures and Tables

**Figure 1 insects-12-00138-f001:**
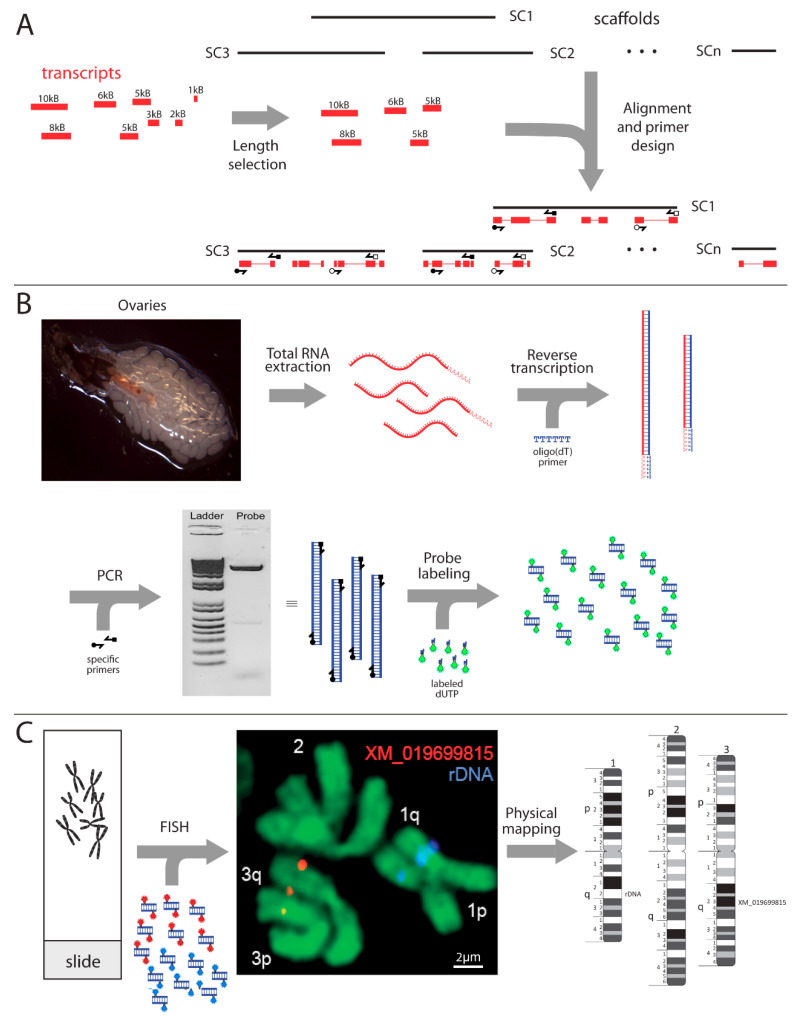
Genome mapping approach based on complementary DNA (cDNA). Procedure includes: (**A**) primer design using transcripts from the *Ae. albopictus* genome; (**B**) extraction of RNA from female ovaries, obtaining complementary DNA based on RNA, amplifying the probe by PCR and probe labeling; (**C**) obtaining the chromosome preparations from mosquito imaginal discs, fluorescent in situ hybridization (FISH), and physical mapping of the probes onto chromosome idiograms. Chromosomes and chromosome arms are indicated by the numbers 1, 2, and 3 and the letters p (petite) and q (long), respectively. A transcript number XM_019699815 is indicated by the same color as signal on FISH image. rDNA stands for ribosomal DNA.

**Figure 2 insects-12-00138-f002:**
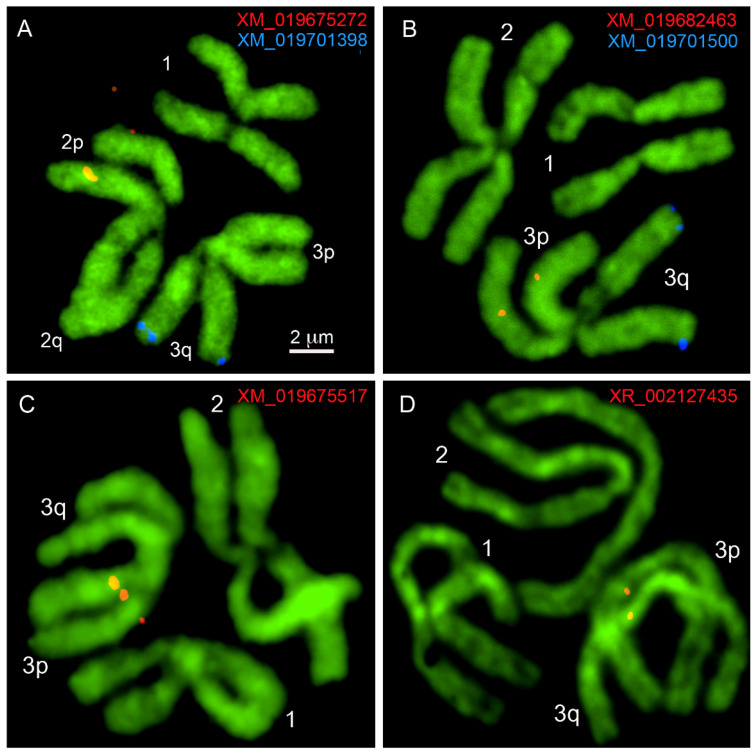
Examples of FISH in mitotic chromosomes of *Ae. albopictus.* The locations of transcripts XM_019675272 and XM_019701398 (**A**); XM_019682463 and XM_019701500 (**B**); XM_019675517 (**C**); and XR_002127435 (**D**) are shown. Chromosomes and chromosome arms are indicated by the numbers 1, 2, and 3 and the letters p and q, respectively. Transcripts numbers on the figure are shown with the same color as the probe signals on the chromosomes.

**Figure 3 insects-12-00138-f003:**
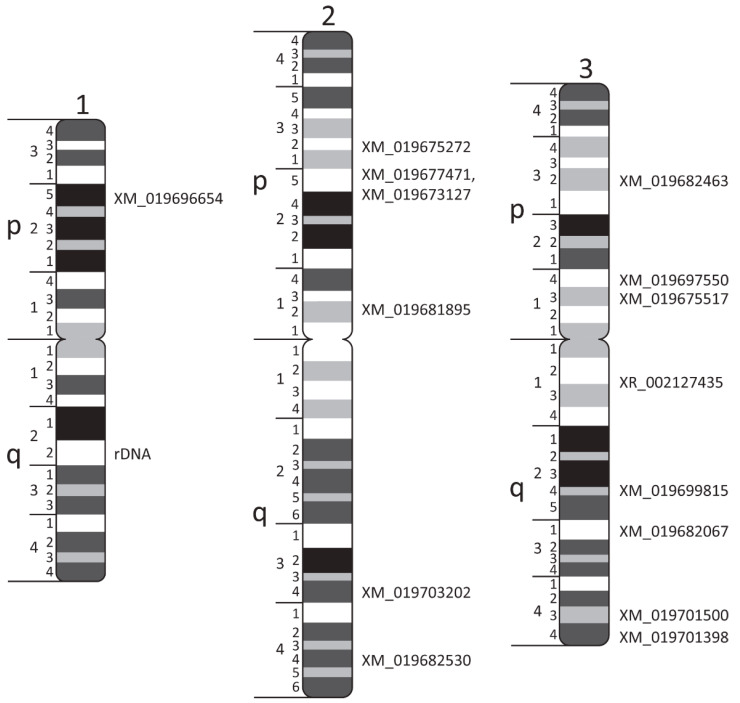
The locations of 15 DNA probes in the chromosome idiograms of *Ae. albopictus*. Chromosomes and chromosome arms are indicated by the numbers 1, 2, and 3 and the letters p and q, respectively. Chromosome divisions and subdivisions are shown on the left side of the idiograms. Transcripts are indicated on the right side of the idiograms; rDNA stands for ribosomal locus.

**Table 1 insects-12-00138-t001:** Transcript locations in the *Aedes albopictus* genome assembly and chromosome map. Probes used for fluorescence in situ hybridization (FISH) are shown by transcript and gene IDs. Probe locations are indicated by scaffold number, chromosome band in the genome, or in the chromosomes.

Transcript ID	Scaffold ID/Number	Chromosome Band	Forward (F) and Reverse (R) Primers
18 S rDNA		1q22	F: ATGCAAAATGCAGGAACCTC R: GGTAATAGCAGCTGGGCTTG
XM_019696654	NW_021838798.1/6	1p25	F: TCGTTCGTGTAGATAAAGTCCAG R: ATGGTTATGAGGTTCCAACAACT
XM_019681895	NW_021838576.1/4	2p12	F: GAAGTAACGGGCTCAGTTCTGGTTTC R: CTTCGAGTAGTTGGACCAGTTCGAGA
XM_019677471	NW_021838576.1/4	2p25	F: AGCTCAACCAAAGGAAGGATTTA R: TGATTGTTCACCTTGTTTTCCAC
XM_019673127	NW_021838576.1/4	2p25	F: AAGTGTGCGGAAAATTGTTTACA R: TAATGGATTGAAGCTGCTTTTCG
XM_019675272	NW_021838153.1/2	2p32	F: TCCCTCTTTTATGAACAGCTGTT R: ACAAAACATTCATGCAGTTGTCA
XM_019703202	NW_021838687.1/5	2q34	F: AAACGACAAGAAATGTTCTGCAA R: TTGTGCCGCATTATCATTCATTT
XM_019682530	NW_021838153.1/2	2q44	F: AACAGAGCGGTATCTACAAAGAG R: GTAGAACACGAAGGCATTAGGTA
XM_019675517	NW_021838832.1/63	3p13	F: ACTTCGGTTATGGGTAAGGTTTT R: CAAAGACATGGGATTTTCTCGTC
XM_019697550	NW_021838154.1/20	3p14	F: GGAAGTTTTGTGTCGAAGGAAAA R: GTCGTCCAGATTGTACAGATCTT
XM_019682463	NW_021837045.1/1	3p32	F: GAGACCAACGCAGAGTACGTCTTCAC R: CGCATAGGCTCTGATGAACTTAGTCG
XR_002127435	NW_017857621.1/18	3q12	F: CCATTAAAAACGCCATCTAGCAA R: TATGAGTGTAGTGTGCTAGCAAG
XM_019682067	NW_021838465.1/3	3q23	F: AGGTACCGTACAAAAGAAGTGAA R: ACGGAACTAAGAAACAAAGTCCT
XM_019699815	NW_021837489.1/14	3q23	F: TTCGGTGGAAAAATCGTTTTGAA R: CGAGTTTCTCTATTGCCACTTTG
XM_019701500	NW_021838465.1/3	3q43	F: CGATGGACTGTTCTTCCAATCTA R: TTTGATTTGTGTTTGTCCCAGAC
XM_019701398	NW_021838465.1/3	3q44	F: TCCCGTTACTTCTACGAAATGAG R: CCATCTTCTGGTTTGCATAACAG

**Table 2 insects-12-00138-t002:** Comparison of the Bacterial Artificial Chromosome (BAC)-based approach and the gene-based method for physical genome mapping.

Comparison Parameters	BAC-Based Genome Mapping	Gene-Based Genome Mapping
BAC library development and BAC-end sequencing	Required	Not required
Primer design	Not required	Required
Blocking of unspecific hybridization with C_o_t_1_ DNA	Required	Not required
Multiple FISH signals are produced in the chromosomes	Often	Never
Additional DNA denaturation step in 70% formamide at 72 °C	Required	Not required
Expensive	Yes	No

## Data Availability

Transcript sequences are avlible at National Center for Biotechnology Information, www.ncbi.nlm.nih.gov.

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
