# Peer review of "A Gene-Based Method for Cytogenetic Mapping of Repeat-Rich Mosquito Genomes"

_insects, 2021, doi:10.3390/insects12020138_

Round 1
Reviewer 1 Report
Review on the manuscript “A gene-based method for cytogenetic mapping of repeat-rich mosquito genomes” by Masri et al.
Summary: This study introduces a method to visualize genomic loci using probes made from transcripts to be utilized for construction of physical mapping of genomes that contain highly repetitive sequences. The method seems to work on Ae. albopictus, approach is interesting and sounds quite simple. However, it seems to lack direct or comparative evidence(s) that it is “simple and robust” compared to the conventional methods, in other words, it is not clear what is scientifically new in this paper as it stands. Addition of description between conventional vs this gene-based FISH (perhaps a figure describing pros/cons of two methods)—to show it is “simpler” than conventional method and more examples (using other species, preferably)—to show its robustness should make this paper much stronger and attractive to the researchers work in the same field. Also, some minor parts need to be clarified.
Points:
Line 92: “C6/C3” C6/36?
Line 93: It is not clear the reason for using C6/36 cell transcriptome instead of the gene model of AaloF1.2, even though the authors performed experiments on Ae. albopictus mosquitoes, not the cell line. Perhaps it has something to do with Fig 1A, but not well explained.
Line 92-99: Can any probe span exon-intron-exon junction (looks so from the figure)? If so, what is a rationale that such probes without an intron (or multiple introns) would specifically bind to a target locus of the genome that contains introns (only a part of probe may hybridize to the genome)?
Also, are there more specific tips for selection of genes (such as minimal length of a single exon to obtain stable and specific hybridization)?
I think that these are the major points for this method and should be further highlighted in the manuscript.
Line 132, 154: Typically, magnification values are for eyepieces (40x objective with 10x eyepiece = 400x). But for imaging, is it the same? Is there 10x lens between 40x or 100x objective and the sensor of the camera?
Line 135: any reasons for this time change?
Line 163: Please use “bases” or “nt” instead of “bp” as transcripts are single-stranded.
Table 1: Please add description in the legend that probe IDs in the figures are the last 4 digits of the Transcript ID or add a column to indicate probe IDs.
Could there be negative control probes to show there is no unspecific hybridization and positive control probes (maybe targeting repetitive sequences) to show the protocol is working?
Is there any reason for not including the FISH results for all the probes, even in the text?
Author Response
Please, see attachment.

Reviewer 2 Report
Physical mapping of single genes is challenging and any improvement is highly welcome. The authors present a new, very clever approach to overcome limitations of this technique. I found the procedure well designed, the description well written and the results astonishing. Therefore, to me, it is a very good contribution to insect cytogenetics. Maybe, the authors can supplment the discussion with summarization/comparison of other single gene mapping in insect, such as tyramide signal amplification (BMC Genet. 2014; 15(Suppl 2): S15).
Author Response
Please see attachment for the reviewers 1 and 2 here.

Reviewer 3 Report
The authors present methodology for developing a physical map of aedine genomes and increase the quality of their genome maps by stitching together genomic scaffolds and tying them to specific chromosomes and chromosomal bands. This work represents an optimization of physical mapping approaches using 15 unique protein-coding gene transcripts. The Sharakhova group has been at the fore front of physical mapping using FISH (DNA probe hybridization) analysis for cytogenetic mapping for several years. This work is important because aedine mosquitoes lack usable polytene chromosomes (unlike drosophila) and their genomes are relatively large due to extensive repetitive regions in the genome leading to poor assembly of the scaffolds and the genome. The paper describes methods for improving genome assembly. Aedes aegypti and Aedes albopictus are important vectors of arboviral disease and an accurate assembly of their genomes is needed. The methods in this paper are straightforward and could be used to assemble genomes of other vectors/insects.
Minor change: Figure 1A ....Alignment misspelled
Minor change: Figures 2 and 3 legends. For the uninitiated readers define p (petit) and q (long) regions of chromosomes and importance of such designation.
Author Response
Please see attachement.

Reviewer 4 Report
The manuscript describes a method for fluorescent in situ hybridization FISH for the cytogenetics of a Culicidae mosquito Aedes albopictus. The FISH technique was used for physical mapping of the genome assisting the genome assembly. Authors emphasized that the use of long sized cDNA for the probe generation for high sensitivity and avoiding repeated genome sequences. The manuscript is generally well written.
However, it is not clear what additional new information than regular FISH is added in the manuscript. Except the empirical suggestions using longer probe, every other procedure is same as regular FISH that has been published by the authors in earlier publications and others. Even the case of improved sensitivity using the long cDNA can be achieved by other numerous FISH tools recently developed. Overall, the manuscript lacks new information.
Round 2
Reviewer 1 Report
Review on the manuscript “A gene-based method for cytogenetic mapping of repeat-rich mosquito genomes” by Masri et al.
Summary: The review of the revised manuscript stands the same as the first review, because my major critique is that it only shows the proposed method works on one species, Ae. albopictus, which makes the method not easily reliable in applying other species. The authors mentioned in the letter, the results have been published elsewhere, which also indicates that results shown in this manuscript are not new finding. It is described in the “Research and Publication Ethics” of the journal that reads:
“Original research results must be novel and not previously published, including being previously published in another language.”
Evidence of this method applied on a second species, will resolve this. Otherwise, it is difficult to support this manuscript as a novel scientific finding. Without providing results from other repeat-rich genomes, I am afraid that I cannot recommend this for publication. The authors should address this before addressing the minor points below if they will resubmit the manuscript.
Alternatively, the authors may consider submitting this in BioProtocol or some other journals that publish protocols for published results.
Some minor points:
In the letter from the authors, they mentioned that the probes are fragmented during nick translation for labelling. The Authors should describe this in the text as the manuscript in the current form gives an impression that the probes are 3800-5000 nt long molecules.
Line 134, 154: Typically, such magnification values are for the view through eyepieces (40x objective with 10x eyepiece = 400x). But for imaging, is it the same? Is there 10x lens between 40x or 100x objective and the sensor of the camera? Description of the magnification of objective would avoid confusion.
throughout the manuscript: Please use “bases” or “nt” instead of “bp” if it refers to single-stranded nucleotides.
Figure 1 and figure 2: need scale bars for the micrographs.
Figure 2: it is still unclear what those 4-digit numbers refer to.
Author Response
Please, see attachment.

Reviewer 4 Report
I find that the authors have tried to respond to the reviewer's points. I think the manuscript yet limited for the novelty and new information.
Author Response
Please, see attachment.

Round 3
Reviewer 1 Report
Review on the manuscript “A gene-based method for cytogenetic mapping of repeat-rich mosquito genomes” by Masri et al.
Before any comments, I genuinely appreciate authors efforts and time for revising the manuscript.
Nonetheless, this round of review stands the same as my second review. Here are the reasons:
- There is no description that DMPI Insects publish articles with previously published results, but not yet been detailed methodologically as article type “Technical Notes”. I have inquired the Assistant Editor for clarification on this matter, but I received no reply.
- I do not have authority to overrule (or violate) the journal’s policy: “Original research results must be novel and not previously published, including being previously published in another language.” to recommend publication.
- Authors appeared not prefer to include the results from another repeat-rich species to overcome this issue.
I think that this work is in a high quality and worth peers’ attention, but it just did not meet the journal’s standard.
I may not further review this manuscript until the authors or editor/journal clear this issue, such as showing an explicit description that the Insects publish detailed methods with published results as “Technical Notes” and/or inclusion of unpublished results.